# Natural Killer Cells and T Cells in Hepatocellular Carcinoma and Viral Hepatitis: Current Status and Perspectives for Future Immunotherapeutic Approaches

**DOI:** 10.3390/cells10061332

**Published:** 2021-05-28

**Authors:** Suresh Gopi Kalathil, Yasmin Thanavala

**Affiliations:** Roswell Park Comprehensive Cancer Center, Department of Immunology, Elm and Carlton Streets, Buffalo, NY 14263, USA; kalathil.suresh@roswellpark.org

**Keywords:** hepatocellular carcinoma, NK cell, CTL, viral hepatitis, immunotherapy

## Abstract

Natural killer (NK) cells account for 25–50% of the total number of hepatic lymphocytes, which implicates that NK cells play an important role in liver immunity. The frequencies of both circulating and tumor infiltrating NK cells are positively correlated with survival benefit in hepatocellular cancer (HCC) and have prognostic implications, which suggests that functional impairment in NK cells and HCC progression are strongly associated. In HCC, T cell exhaustion is accompanied by the interaction between immune checkpoint ligands and their receptors on tumor cells and antigen presenting cells (APC). Immune checkpoint inhibitors (ICIs) have been shown to interfere with this interaction and have altered the therapeutic landscape of multiple cancer types including HCC. Immunotherapy with check-point inhibitors, aimed at rescuing T-cells from exhaustion, has been applied as first-line therapy for HCC. NK cells are the first line effectors in viral hepatitis and play an important role by directly eliminating virus infected cells or by activating antigen specific T cells through IFN-γ production. Furthermore, chimeric antigen receptor (CAR)-engineered NK cells and T cells offer unique opportunities to create CAR-NK with multiple specificities learning from the experience gained with CAR-T cells with potentially less adverse effects. This review focus on the abnormalities of NK cells, T cells, and their functional impairment in patients with chronic viral hepatitis, which contributes to progression to hepatic malignancy. Furthermore, we discuss and summarize recent advances in the NK cell and T cell based immunotherapeutic approaches in HCC.

## 1. Introduction

HCC is one of the leading causes of cancer-related death globally [1]. The major risk factors causing HCC include chronic viral infection, alcohol-related cirrhosis, and nonalcoholic steatohepatitis (NASH) [2]. Treatments for HCC include hepatectomy, liver transplant, radiofrequency ablation (RFA), hepatic transarterial chemoembolization (TACE), chemotherapy, and molecular targeted therapy [3]. However, these therapeutics are not effective for advanced forms of HCC and the risk of recurrence is very high in these patients. This challenging clinical scenario warrants novel efficacious and life-prolonging therapeutic strategies for patients with HCC. Compelling evidence suggests that NK cells play a central role in the immune function of the liver and in the immune defenses against HCC, indicating that HCC might be an ideal candidate for NK cell-based immunotherapeutic approaches [4,5]. The HCC tumor microenvironment (TME) is characterized by a severe dysfunction of the immune system through multiple mechanisms, including accumulation of various immunosuppressive factors, recruitment of regulatory T cells (Tregs) and myeloid-derived suppressor cells (MDSC), and induction of T cell exhaustion followed by the interaction between immune checkpoint ligands and receptors [6,7].

Hepatitis B virus (HBV) and hepatitis C virus (HCV) are major health problems worldwide, with over 300 [8] and 170 million people infected, respectively [9]. Both HBV (DNA virus) and HCV (RNA virus) are hepatotropic [10]. HBV and HCV can establish a chronic persistent infection in many exposed individuals, and the host immune response is believed to play an important role in determining the fate of infection [8,11]. Impaired activity of NK cells has been proposed as a mechanism contributing to viral persistence and chronicity of HCV infection [11]. In chronic HBV infection and HCV infection the frequencies of NK cells and their ability to produce proinflammatory cytokines such as IFN-γ and TNF-α are significantly reduced [11,12]. Chronic HBV and HCV infections are the major viral etiological factors of HCC [13]. NK cells have both direct and indirect role in anti-viral immunity either by producing cytokines and by exerting cytotoxic functions against virus-infected cells directly or by supporting virus specific T cell responses via IFN-γ production [14,15]. Nevertheless, in chronic HBV infection, NK cells have been described to be more pathogenic than protective with a poor capacity to produce anti-viral cytokines [14]. In addition, NK cells can exert regulatory activity and possibly suppress adaptive immune responses in the setting of persistent viral infection. Therefore, NK cell based therapeutic strategies should be focused not only to improve NK cell killing potential of virus infected cells but to alleviate NK cell mediated suppressive effect on virus specific T cells [14]. Additionally, immune mediated liver damage by TRAIL-expressing NK cells through their interaction with hepatocytes expressing TRAIL-death receptors has been reported in chronic HBV [8].

## 2. NK Cells in Chronic Viral Hepatitis

Chronic HBV and HCV infections are the leading cause for the development of liver cirrhosis and subsequent transformation to HCC [16]. Both HBV and HCV replicate and grow inside the hepatocytes which upregulate ligands of NKG2D, causing NK cell activation. In patients with chronic HBV and HCV infection, low frequencies of circulating NK cells combined with diminished production of proinflammatory cytokines, TNF-α, and IFN-γ have been reported as compared to healthy controls [11,12]. Nevertheless, controversy exists regarding the impact of chronic viral infection on NK cell cytolytic function; while several researchers reported impaired NK cytotoxicity, other reports showed that effector function was not changed [17,18]. The differences in outcomes between these studies were probably due to the lack of standardized protocol and reagents, as well as the heterogeneity of patients. NK cells exert a direct antiviral effect through (1) perforin and tumor necrosis factor-related apoptosis-inducing ligand (TRAIL) dependent cytolysis of infected cells and (2) induction of IFN-γ production [11,12,19]. Phenotypic heterogeneity of NK cells has been reported among HBV and HCV patients with elevated expression levels of activating receptor NKG2D and a decrease in inhibitory receptor expression in chronic HCV patients, while in chronic HBV patients, the percentage of activating NKG2C^+^ NK cells increased and the inhibitory receptor expression was normal [12]. The reduced percentage and impaired function of NK cells in chronic viral hepatitis patients were believed to contribute to the disease progression and the transformation to HCC. In acute HCV infection, hepatic NK cells show an increased expression of NKp46 and potentiality to degranulate and to produce IFN-γ following strong activation by IFN-α/β and other cytokines (IL-12, IL-15, IL-18) [20]. NK cell activation and proliferation in response to HCV, mediated by monocyte-derived cells and the OX40/OX40L axis, suggests cross talk between monocytes and NK cells [21]. NK cell repertoire in HBV patients co-infected with HCV is biased towards memory like phenotype associated with elevated CD16 mediated ADCC effector function, [15]. Even though, liver NK cells maintain their cytotoxic potential in chronic HBV infection via the upregulation of TRAIL [8], they eliminate autologous HBV specific CD8^+^ T cells expressing high levels of death receptor for TRAIL. Hence, this NK cell-mediated depletion of antigen-specific CD8^+^ T cells impairs adaptive antiviral immunity in chronic HBV-infected patients and contributes to viral persistence [22]. In chronic HBV patients, elevated frequency of killer cell lectin-like receptor subfamily G member 1^+^ (KLRG1) NK cells are detected and are implicated to causes reduction in hepatic fibrosis. Harnessing this antifibrotic function may provide a novel therapeutic approach to treat liver fibrosis in patients with chronic HBV [15]. Effective HBV vaccination and anti-HCV drugs have significantly reduced the number of chronic viral hepatitis patients and this will likely reduce the number of HCC cases in the future [23,24].

## 3. Role of T Cells in Viral Hepatitis

Earlier studies have demonstrated the central role of antigen specific CD4^+^ and CD8^+^ T cell responses in protection against viral persistence [9,25]. Potent CD4^+^ helper and CD8^+^ effector T cell responses have been observed in patients with hepatitis [26,27] and experimentally infected chimpanzees that clear HBV and HCV infection [28]. Antigen specific T cell responses can be detectable in the liver and peripheral blood approximately 4 to 8 weeks after infection. HLA- alleles, B27, B57, and A3 show strong association with protective immunity [29,30,31]. Protective T cell responses tend to target epitopes that do not allow virus escape mutations [25]. Even though HBV and HCV adopt different immunological mechanisms for viral persistence, a common feature is the subversion of virus specific T cell immune responses, that is characterized by progressive functional exhaustion and finally deletion of virus specific CD4^+^ and CD8^+^ T cells [32]. CD4^+^ T cells lose their capacity to produce IL-2, a cytokine that is critical for T cell proliferation and this is followed by sequential loss of cytotoxicity and TNFα and IFN-γ production. In addition, intracellular expression of Bcl-2 interacting mediator (Bim), a pro-apoptotic protein increases in virus specific CD8^+^ T cells of HBV or HCV infected patients [33,34]. A shift in the balance between T cell sustaining cytokine IL-2 and anti-inflammatory cytokines IL-10 and TGF-β dampens the virus specific T cell responses of infected patients [35,36]. Viral mutations in dominant T cell epitopes that have been selected to establish chronic infection render the remaining T cell responses irrelevant. These viral escape mutations have been described in both HBV [37] and HCV-infected patients [38], but they are more frequent in HCV infection as a result of the high replicative fitness of HCV.

The metabolic fitness of virus specific CD8^+^ T cells plays a central role in mounting an efficacious anti-viral immune response during chronic HBV or HCV infection. One important mechanism responsible for the failure of intrahepatic HCV-specific CD8^+^ T cells to clear HCV in persistent infection is due to the loss of mitochondrial fitness. In HCV infection, virus-specific CD8^+^ T cells display a marked upregulation of transcription of genes associated with impaired glycolytic and mitochondrial functions, leading to exhaustion of HCV-specific T cells [39]. HBV-specific CD8^+^ T cells are markedly impaired at multiple levels and show substantial downregulation of various cellular processes centered on extensive mitochondrial alterations [40]. Multiple strategies to increase the mitochondrial potential of exhausted virus specific CD8^+^ T cells and thereby rescue them from functional exhaustion has been reported recently. In chronic HBV infection, IL-12 has been shown to revitalize exhausted HBV-specific T cells by reducing their dependence on glycolysis and thereby increasing their mitochondrial potential, which is critical for viral control. Furthermore, IL-12 administration has been reported to significantly decrease the levels of pro-apoptotic protein Bim, which potentially cause attenuation of HBV specific CD8^+^ T cells [41,42]. IL-15 induced upregulation of autophagy in human liver-resident CD8^+^ memory T cells has been described as a protective mechanism against mitochondrial depolarization and to optimize effector T cell function [43]. In chronic HBV patients, IL-2 has been described as a potent immunotherapeutic that can rescue CD8^+^ T cells rendered dysfunctional by intrahepatic priming; this highlights the importance of IL-2-based strategies that should be considered for the treatment of chronic HBV infection [44]. Significant improvement of mitochondrial function with concomitant enhancement in anti-viral CD8^+^ T cell functions was elicited by mitochondrion-targeted antioxidants, which suggests a central role for reactive oxygen species (ROS) in T cell exhaustion [40]. Thus, mitochondria represent promising therapeutic targets to treat chronic HBV infection. Additionally, combinatorial treatment of polyphenolic compounds (resveratrol, oleuropein) and antioxidants improved anti-HBV T cell responses in chronic HBV patients. Therefore, combination of antioxidants and natural polyphenols represents a promising strategy for chronic HBV therapy [45]. GS-9688 (selgantolimod), a toll-like receptor 8 (TLR8) agonist induces cytokines in human PBMC that are able to activate antiviral effector function by multiple immune cells, including HBV-specific CD8^+^ T-cells, T follicular helper (TFh), NK cells, and MAITs [46]. In chronic HCV patients treated with direct-acting antiviral (DAA) therapy, exhausted HCV-specific CD8^+^ T cells remain functionally and metabolically impaired at multiple levels and altered mitochondrial dysfunction exhibited by exhausted HCV-specific CD8^+^ T cells could not be normalized even after HCV clearance in most patients. This finding implies that HCV cure does not lead to protective immunity and that re-infections have frequently been observed [47]. Nevertheless, functional exhaustion of HCV specific CD8^+^ T cells associated with impairment in glycolytic and mitochondrial function can be alleviated by histone methyltransferase inhibitors [39].

Individuals who spontaneously resolve acute HCV infection develop long-lived CD45RA^−^ CD127^+^ CD8^+^ memory T cells and are therefore less likely to become infected upon second exposure [25]. Presence of antigen specific CD8^+^ T cells in the peripheral blood and liver, detection of IFN-γ, CD3, CD4, and CD8 transcripts are associated with a decline in viral load [35]. The magnitude of the CD8 T cell response is a key determinant of spontaneous resolution and several epitopes are recognized simultaneously in humans with acute resolving HCV, while much fewer epitopes are recognized in individuals developing chronic infection [9,35]. Multiple exhaustion markers, such as PD-1 [48],Tim-3 [49], CTLA-4 [50], CD160, KLRG-1, and 2B4 [51], are differentially expressed on CD8^+^ T cells during acute and chronic viral infections, suggesting a spectrum of exhaustion markers that correlate with decline in CD8^+^ T cell function and persistent infection. The decline in CD8^+^ T cell function were reversible in vitro upon blockade of the PD-1, CTLA-4, and/or Tim-3 pathways. Early induction of IL-21 producing Th17 cells is critical to mediate help and limit exhaustion of virus-specific CD8 T cells [25].

In summary, spontaneous resolution of acute viral hepatitis correlates with early emergence of CD127^+^ antigen specific CD8^+^ T cells. Several factors contribute to failure of this response in individuals who become chronically infected. Escape mutations in targeted epitopes facilitate viral evasion of the immune system, loss of CD4^+^ T cell help, or a switch to an immunoregulatory profile further compromises the antiviral capacity of virus specific CD8^+^ T cells. Finally, continued viral replication contributes to exhaustion of HCV-specific T cells through persistent antigenic stimulation leading to progressive loss of function and diminished survival of virus-specific CD8 T cells. Additionally, exhaustion is exacerbated by increased expression of the ligands to the inhibitory receptors on infected hepatocytes from a therapeutic angle, the blockade of these inhibitory receptors combined with selective stimulation of co-stimulatory pathways is being pursued both in vitro and in vivo as a means to rescue exhausted T cells and to restore their function.

## 4. NK Cells and Immune Surveillance

In the normal physiological state, the liver is tolerogenic to immune stimuli from bacterial products, environmental toxins, and food antigens transported via the hepatic portal vein and hepatic artery, thereby avoiding unnecessary tissue damage [52,53]. This immune-tolerant nature of the liver microenvironment contributes to the immune evasion of HCC [54]. NK cells are the predominant innate immune cell subset in the liver and are critical in host defense against invading pathogens, liver injury, and tumor development [55,56]. In humans, almost 50% of intrahepatic lymphocytes are NK cells that are strongly imprinted in a liver-specific manner and play a key role in liver immune responses both under normal homeostasis and pathological conditions [57]. NK cells are considered as the first line of defense against viral infections and tumor invasion [16]. NK cells recognize and kill virus infected cells and tumor cells without major histocompatibility complex restriction or prior sensitization, via cytotoxicity or cytokine secretion [58,59]. NK cells are known for their ability to rapidly recognize and clear viral-infected, tumor-transformed, and stressed cell targets in the absence of antigen specificity [60,61]. NK cells in human are phenotypically defined as CD3^−^CD56^+^ large granular lymphocytes. NK cells express activating receptors such as CD16, NKp30, NKp44, NKp46, NKp80, NKG2D, CD244, CD226; cytokine receptors such as IL-2R, IL-28R, IL-12R, IFNR, IL-15R, IL-18R, IL-1R8, IL-10R, and TGFβR [58,62]. The inhibitory receptors expressed by NK cells include NKG2A, KLRG1, KIRs, TIGIT, TIM3, Siglecs, PD-1, LAG3, A2AR, LAIRs, and ILTs [58,63]. Under homeostatic conditions, human circulating NK cells represent about 5–15% of circulating lymphocytes and are subdivided into two main subsets defined based on their differential expression of CD56 and CD16, namely CD56^bright^ CD16^−^ and CD56^dim^ CD16^+^ [64,65]. Relatively low amounts of intracellular cytotoxic granules, perforin, and granzyme B are the features of CD56^bright^ CD16^neg^ NK cells and these cells have low cytotoxic potential and are not able to perform antibody dependent cellular cytotoxicity (ADCC) mediated killing since they do not express CD16. On the other hand CD56^dim^ NK cells exerts both potent cytotoxicity and ADCC given their high constitutive expression of CD16 in response to activation by IL-12, IL-15, and IL-18 [66]. A growing body of evidence suggests that impaired NK cell function leads to the body’s failure to eliminate tumor cells, which indicates that tumor cells could be killed more effectively through enhancing the activity of dysfunctional NK cells [16,67]. Strong evidence for the role of NK cells in tumor surveillance is derived from correlative studies which demonstrated that a low NK cytotoxicity profile of peripheral blood lymphocytes correlated with an increased risk for cancer [68]. Impairment in NK cell function was observed in many types of liver diseases including chronic viral hepatitis and HCC, causing impediments to elimination of virus infected cells or tumor cells efficiently. HCC tumor tissues were markedly infiltrated by activated NK cells and Tregs, which should be considered as candidate therapeutic targets in HCC multidisciplinary treatments [69].

The functions of NK cells are strictly regulated by the balance of activating receptors and inhibitory receptors interacting with target cells [63]. These receptors can bind to specific ligands, mainly the major histocompatibility complex class (MHC-1) that is expressed on healthy hepatocytes, which interacts with inhibitory receptors on NK cells and prevents the activation of NK cells. The inhibitory Killer Ig-like receptors (KIRs) that recognize classical MHC-I alleles and the C-type lectin like receptor NKG2A that forms a heterodimer with the CD94 molecule (CD94/NKG2A) and bind HLA-E, a non-classical MHC-I molecule prevent NK cell activation (Figure 1A) [70]. During viral infection, hepatocytes upregulate stimulatory ligands for NK cell activating receptor NKG2D and as a result activation signal surpasses signaling through the inhibitory receptors KIRS and NKG2A, resulting in cytokine release and cytolysis of target cells (Figure 1B). Either the decreased expression of, or the absence of MHC-I on target cells triggers NK cell killing, a phenomenon known as “missing-self hypothesis”, via the engagement of activating NK receptors (Figure 1C) [71] which bind their putative ligands expressed on virus-infected, malignant, or stressed cells. NK cells can thus directly eradicate infected cells or tumor cells lacking MHC-1 molecule expression [55]. Therefore, approaches to regulate the balance between activating and inhibitory receptors in NK cells, might also lead to the successful treatment for various liver diseases including HCC [70,72]. Hepatocarcinoma cells express ligands of several activating NK receptors (NKR), including NKp30, natural killer receptor group 2, member D (NKG2D), and DNAM-1, such as the B7 protein homolog 6, the major histocompatibility complex class I chain-related protein A and B (MICA/B), and CD155, respectively, whose expression can correlate with HCC outcome [73,74]. Among these receptors, NKG2D/NKG2DL axis plays a pivotal role in the detection and elimination of virus infected cells and tumor cells. Nonetheless, viruses and tumor cells devise numerous strategies to evade detection by NKG2D immunosurveillance system [75]. NKG2D expression is down-regulated by transforming growth factor β (TGF-β), produced in the context of tumor development, which induces immune tolerance and controls inflammation via a suppressive action on innate and adaptive immunity [76,77]. Therefore, targeting TGF-β appears to represent an intervention strategy to boost NK cell mediated tumor immunity and should be considered in future cancer treatment modalities. The importance of NK cells and their activating receptor/ligand axis in HCC immune surveillance has been extensively studied and patients with decreased expression of major histocompatibility complex class I chain-related protein A (MICA) in HCC tissue showed reduced disease-free and overall survival compared with patients with preserved MICA expression [78]. Innate lymphoid cell (ILC-1) or liver-resident NK cells play versatile roles in liver diseases. Liver-resident NK cells are CD56^bright^Eomes^high^Tbet^low^Hobit^+^TIGIT^+^CD69^+^CXCR6^+^CD49e^−^, express higher levels of NKG2D, NKp46, TRAIL, and Fas ligand and possess cytotoxicity against HCC cells [79]. However, during tumor progression liver resident NK cells in tumor tissue down regulate NKG2D [80] and express inhibitory receptors PD-1, CD96, and TIGIT and are rapidly exhausted [81]. Thus, NK cells in the tumor tissues of patients with advanced HCC are dysfunctional or in an exhausted state [5], suggesting that NK-cell exhaustion may be contributing to HCC progression [72].

## 5. NK Cells-Based Immunotherapy of HCC

The exhaustion of NK cells as reflected by diminished cytotoxicity and impaired cytokine production may serve as a predictive and prognostic marker of HCC [5]. The absolute number of circulating NK cells and intrahepatic NK cells are positively correlated with HCC patients’ survival and prognosis [5,16]. In humans, both intra and extra-hepatic NK cells represent critical cells of our innate immune system, and play a key role in the body’s immune responses against cells infected with HBV or HCV and development of HCC [82]. Therapeutic interventions based on the activating receptor/ligand axis and/or IL-15 stimulation of effector NK cells represent a novel approach in cell based immunotherapy of HCC [73]. NK cells target tumor cells sensitized by monoclonal antibodies specific for tumor associated antigens GPC-3 or AFFP on tumor cells by mediating ADCC (Figure 1D) and this would also be a promising therapeutic strategy after activation of NK cells with cytokines such as IL-15 [73,83]. Activated NK cells generated from the peripheral blood of healthy donors using the K562-mb15-41BBL cell line as a stimulus, exerted remarkably high cytotoxicity against HCC cell lines, that was significantly higher than that of unstimulated or IL-2 activated NK cells. These expanded allogeneic or autologous NK cells, after genetic modification with NKG2D-CD3ζ-DAP10 chimeric receptor were shown to have enhanced anti-HCC cytotoxicity. These results warrant clinical trial of expanded allogeneic or autologous NK-cell infusions in patients with HCC, possibly after genetic modification with NKG2D-CD3ζ-DAP10 [84]. Cryoablation combined with allogeneic NK cell therapy has been shown to markedly improve the progression free survival of patients with advanced HCC [85]. A recent report on Tim-3-mediated dysfunction of both tumor-infiltrating, liver resident NK cells and conventional NK cells, provides a new insight into immune checkpoint based targeting for NK cell for immunotherapeutic intervention in HCC patients [86]. Accumulation of CD49a^+^ NK cells in the human liver indicates deteriorating disease condition and poor prognosis and thus elimination of CD49a^+^ NK cells could represent a potential therapeutic strategy in the immunotherapy of HCC [81]. Human intratumoral CD96^+^ NK cells represent functionally exhausted NK cells and patients with elevated levels of intratumoral CD96 expression exhibit poorer clinical outcomes. Therefore, reversing NK cell exhaustion by blocking the interaction between CD96 and its ligand CD155 might have therapeutic potential in the treatment of HCC patients [72]. NK cell activity is significantly improved when KIRs of donor and HLA class I of the recipient are incompatible, and this results in alleviation of inhibitory signals. Thus, KIR-ligand incompatibility seems to be critical for the enhanced efficacy of allogeneic NK cell [16]. Irreversible electroporation of tumor cells combined with allogeneic NK cell immunotherapy significantly increases the overall survival of patients with stage IV HCC and this combinatorial approach had synergistic effect by reduction of AFP levels with clinical benefit [87]. Dose dependent administration of sorafenib synergizes with NK cell function and therefore combinatorial therapy with sorafenib and NK cells may also improve the outcome of applied therapeutic approaches for HCC patients [88]. Trifunctional NK cell engagers (NKCEs), targeting two activating receptors, NKp46 and CD16, on NK cells and a tumor antigen on cancer cells trigger tumor cell destruction and this approach was superior to therapeutic antibodies both in vivo and in vitro [89]. An anti-NKG2A monoclonal antibody that binds NKG2A, the inhibitory receptor present on both NK cells and CD8^+^ T cells, promoted anti-tumor immunity of these immune cell subsets and thus, anti-NKG2A might have better therapeutic efficacy since it targets both NK cells and CD8^+^ T cells [90,91]. Importance of CD6 in NK cell activation has hitherto not been appreciated and recently an anti-CD6 monoclonal antibody, UMCD6 has been shown to upregulate the expression of the activating receptor NKG2D with concomitant downregulation of the inhibitory receptor NKG2A on both NK cells and CD8^+^ T cells, with concurrent increases in perforin and granzyme B production (Figure 1D) [92]. The dual capability of anti-CD6 monoclonal antibody to enhance killing of cancer cells through its effects on both CD8^+^ T cells and NK cells opens a new avenue for cancer immunotherapy [92]. Tumor infiltrating NK cells co-expressing CD49a and Eomes with reduced cytotoxic potential, have been reported in HCC and this NK cell subset was implicated to have proangiogenic function [93].

Metabolic fitness of NK cells has a crucial role in tumor surveillance. Fragmentation of mitochondria in liver NK cells of HCC patients was correlated with reduced cytotoxicity and NK cell loss, resulting in tumor evasion from NK cell surveillance, with predicted poor survival in HCC patients. Inhibition of mitochondrial fragmentation improved mitochondrial metabolism, survival, and the antitumor capacity of NK cells [94]. In colorectal liver metastasis (CRLM), liver-resident NK cells infiltrating into the tumors were unable to regulate intracellular pH resulting in mitochondrial stress and apoptosis. Hence, targeting CRLM metabolism represents a novel therapeutic strategy to restoring local NK-cell activity and preventing tumor growth [95] Functional rescue of NK-cells in solid tumors represents a central aim for new immunotherapeutic strategies. Metabolic defect and functional impairment of circulating NK-cells in patients with HCC could be attributed to TGF-β. Incubation of NK-cells from healthy donors with TGF-β-rich plasma from HCC patients can recapitulate the metabolic and functional impairment predicted by phenotypic and gene expression profiles of HCC patients. TGF-β antagonism could partially restore functional defects of NK-cells, suggesting that TGF-β among other molecules may represent a suitable target for immunotherapeutic intervention aimed at functional restoration of NK-cell. [96]. These studies highlight the potential mechanism of immune escape through mitochondrial dysfunction that might be targetable and could reinvigorate NK cell-based cancer treatments.

Potential advantages that NK cell therapeutics have over T cell based therapies include more manageable safety profiles and no requirement for autologous cells for adoptive transfer [97]. Accumulating evidence indicates that harnessing NK anti-tumor immunity represents a potentially powerful therapeutic approach for HCC, given the technical advancement in the activation and expansion of NK cells.

## 6. CAR-NK Cells

Several strategies using genetic modification techniques have been developed to improve the targetability and efficacy of NK cell cytotoxicity to tumor cells. The strategy used for generating CAR T cells has also been applied to NK cells, improving the specificity and efficacy of NK cell therapy [98,99]. CAR-NK cells are primary NK cells genetically engineered to express chimeric antigen receptor (CAR), such as AFP or GPC-3, which facilitate binding of relevant antigen in a specific manner. Additionally, NK cells express receptors for Fc portion of immunoglobulin (FcR) which can bind sensitized target cells and mediate cytolysis of NK cell susceptible target cells (ADCC). CAR-NK cells have been reported to reduce the risks of autoimmune responses and neoplastic transformation because they have a shorter lifespan than CAR T cells [98,100]. In addition, cytokines released from NK cells, such as IFN-γ and granulocyte-macrophage colony-stimulating factor (GM-CSF), are considered safer than the cytokine storm that results from CAR T cell therapy [98,100]. Moreover, ADCC is an additional NK cell-mediated tumor-killing strategy that could augment CAR-NK antitumor activity. Besides autologous and allogeneic NK cells, NK-92 cells, an NK cell line, is also used in clinical trials of cancer therapy, with promising results observed in sarcoma and leukemia patients [101]. Among genetically modified NK cells, glypican-3 (GPC3) specific CAR-NK-92 cells were reported to have high antitumor activity against HCC xenografts expressing both low and high levels of GPC3. The specificity of GPC3 CAR-NK-92 cells was confirmed by potent cytolytic activity displayed in vitro cytotoxicity assay against GPC3^+^ HCC cells, while they were not cytotoxic to GPC3^−^ HCC cells [102]. Thus, GPC3-specific CAR-NK cells represent a novel treatment option for GPC-3^+^ HCC patients (Figure 1D). Granule polarization is a critical step leading to NK cell mediated cytolysis of resistant NK cell target, MDA-MB-453 breast cancer cells unmodified NK cells create conjugates with this resistant cancer cells and respond by granule clustering but fail to induce granule polarization and consequent release of lytic enzymes. Nevertheless, retargeting by ErbB2-targeted CAR-expressing NK cell or high affinity FcR transgenic NK cells plus monoclonal antibody Herceptin (ADCC) provides signals necessary to achieve granule polarization and tumor cell lysis, thereby circumventing tumor cell resistance [103].

## 7. T Cells-Based Immunotherapy of HCC

Even though, HCC is considered as a poorly immunogenic tumor, studies have shown that HCC patients who have a high level of lymphocyte infiltration in their tumors or high frequencies of circulating antigen specific CTL have a lower risk of recurrence and a better prognosis [104,105]. These findings confirm that HCC patients develop T cell mediated anti-tumor immunity that can suppress tumor progression. Several studies in the past decade identified different tumor associated antigens (TAAs) and their respective T cell epitopes [106,107,108,109,110,111]. Among the HCC TAAs, the immune response to α-fetoprotein (AFP) has been studied extensively since CTL epitopes for AFP were identified at an early stage of tumorigenesis [112,113]. Earlier studies have demonstrated that AFP-epitope specific T cell are more frequently seen in HCC patients than in healthy individuals and the ratio of these T cells in peripheral blood increases after radiofrequency ablation and TACE [105,107]. Human T cells were transduced with mouse TCR specific for HLA-A2/AFP complex and these engineered human T cells have been reported to specifically recognize HLA-A2^+^ AFP^+^ HepG2 tumor cells and produce effector cytokines. Importantly, these TCR gene engineered T cells could specifically kill HLA-A2^+^ AFP^+^ HepG2 tumor cells, sparing normal hepatocytes in vitro [114]. AFP_2-11_ -HLA-A*24:02 specific TCR transfected T cells could specifically activate and kill AFP_2-11_ pulsed T2-A24 cells and AFP^+^ HLA-A*24:02^+^ tumor cell lines, demonstrating that AFP_2-11_ epitope can be naturally presented on the surface of AFP^+^ tumor cell lines [115]. Antigens that are specific for HCC such as GPC3 and AFP are being tested as part of CAR constructs [116,117].

In addition to AFP, several TAAs have been identified for HCC that include human telomerase reverse transcriptase (hTERT), melanoma antigen gene-A (MAGE-A), glypican-3 (GPC3), and NY-ESO-1 [108,118,119]. The ratio of TAA-specific CTLs in PBMCs of HCC patients varies from 10 to 60.5 cells/300,000 PMBCs, and only 3–19% of patients had CTLs specific to the epitopes derived from different antigens [120]. Nonetheless, RFA enhances TAA-specific T cell responses and the number of TAA-specific T cells induced is associated with recurrence-free survival of HCC patients. To maintain the TAA-specific T cell responses induced by RFA and to improve the immunological outcome for HCC, additional combinatorial strategies such as vaccination or immunomodulatory drugs might be useful [120]. Peptides derived from squamous cell carcinoma antigen when recognized by T cells (SART-3) elicited peptide specific CTL IFN-γ responses upon stimulation of PBMC from HCC patients and were shown to mediate cytolysis of HCC cells expressing the antigen. Vaccination with SART-3 derived peptides resulted in the infiltration of antigen-specific CTL producing IFN-γ into the tumor site [111]. Even though normal mucosal associated invariant T (MAIT) cells could induce apoptosis of HCC cells, they are exhausted and functionally impaired in TME. In the context of liver TME, tumor-derived MAIT cells secreted less IFN-γ, IL-17, and granzyme B, perforin, but they secreted more IL-8, which promotes tumor angiogenesis and progression. Therefore, high density of tumor infiltrating MAIT cells have been reported to indicate disease progression and poorer outcome of HCC and these cells potentially could serve as a novel therapeutic target in T cells-based immunotherapy of liver cancer [121,122].

Adoptive cell transfer techniques, lymphokine-activated killer (LAK) and tumor-specific CTL therapy have been evaluated in HCC patients, and tumor-specific CTL therapy was reported to be more effective in terms of disease outcome [123]. LAK cells administered as an adjuvant to surgery resulted in a 5-year recurrence-free survival rate of 38% HCC patients as compared to 22% patients who did not receive LAK cells [124]. Cytokine induced killer (CIK) cells isolated from PMBCs of patients and cultured with a cytokine cocktail were shown to have highly potent antitumor activity [125]. Furthermore, CIK cell therapy improved the overall survival (OS) of patients when used in combination with either RFA or TACE [125,126]. Ablative therapies induce a peripheral immune response which may enhance the effect of anti-CTLA-4 treatment in patients with advanced hepatocellular carcinoma. Tremelimumab in combination with tumor ablation represents a potentially novel treatment modality for patients with advanced HCC, that leads to the accumulation of intratumoral CD8^+^ T cells in conjunction with reduction in HCV viral load [127]. Microwave ablation in HCC patients leads to abscopal effects in distant lesions reflected by enhanced tumor antigen specific T cell IFN-γ production that correlated with long term survival. These studies provide additional evidence for the potential synergistic effect that could be achieved by combination of local ablation and immunotherapeutic strategies against this challenging disease [128]. The above studies indicate that immune cell therapy is effective in reducing the recurrence rate, which is typically high for HCC patients following curative treatment.

Effector CD8^+^ T cells are in an exhausted state in advanced HCC patients, caused by persistent antigen stimulation, local release of IFN-γ and immunosuppressive factors [129,130]. Elevated expression levels of inhibitory receptors such as programmed cell death protein 1 (PD-1), lymphocyte activation gene-3 (LAG-3), 2B4 (CD244) T cell immunoglobulin, and mucin domain 3 (TIM-3), cytotoxic T lymphocyte-associated antigen 4 (CTLA-4) have been reported in exhausted T cells [131,132]. These inhibitory receptors are druggable targets for tumor immunotherapy [131,132]. Recently, checkpoint inhibitors have been used successfully in cancer treatment; however, they are only effective in 10–40% of cases, and some cancers are resistant to checkpoint inhibitors. PD-L1 expression on intratumoral hepatic stellate cells or peritumoral neutrophils also contributes to the impairment of T cell mediated anti-HCC immunity [133,134]. Blockade of PD-1-PD-L1 axis interaction, therefore, offers a promising strategy to reinvigorate exhausted CD8^+^ T cells and this has been confirmed in clinical trials demonstrating durable objective responses in advanced HCC patients treated with nivolumab or pembrolizumab [135,136,137]. Nevertheless, phase III trials did not reveal any statistically significant improvement in survival benefit [138]. A recent phase-I/II/III study reported that the combination of atezolizumab and bevacizumab resulted in a 62% response rate in advanced HCC patients, which has been approved as a first-line treatment for patients with advanced or metastatic HCC [139,140,141]. Radiotherapy has been shown to enhance immunogenicity of tumors through various mechanisms of immune modulation and therefore combination of ICIs and RT is being evaluated as an approach for HCC to take advantage of the synergistic effect [142]. TIGIT is enriched in PD1^high^CD8^+^TILs of HCC patients, and this subset represents the most dysfunctional and exhausted CD8^+^TIL phenotype and dual blockade of TIGIT and PD-1 improved the cytotoxic potential of CD8^+^ TIL in HCC patients [143]. The above results indicate that combinatorial strategies are more efficacious than CPI monotherapy and new therapeutic approaches are underway to determine whether the addition of tyrosine kinase inhibition, the anti-vascular endothelial growth factor antibody bevacizumab, or additional immune CPIs can augment the proportion of patients with response to PD-1 or PD-L1 inhibition.

Genetically modified autologous T cells that express an HBsAg specific T cell receptor, mediated a reduction in HBsAg levels in HBV related HCC without exacerbation of liver inflammation or “off-target” toxicities [144]. This is the first human proof of principle study for TCR redirected therapy against HBV associated hepatic malignancies. GPC3-CAR T cells have been shown to eliminate GPC3^+^HCC cells and tumors in a patient derived xenograft model [145]. However, dual-targeted CAR-T directed at GPC-3 and asialo-glycoprotein receptor (ASRG1) exerted superior anticancer activity and persistence than single-targeted CAR-T cells in two GPC3^+^ASGR1^+^HCC xenograft models [146]. Thus, T cells carrying two complementary CARs against GPC3 and ASGR1 may reduce the risk of off-tumor toxicity while maintaining relatively potent antitumor activity on GPC3^+^ASGR1^+^ HCC. In order to circumvent T cell exhaustion induced by immune checkpoint expression, an enhanced version of CAR T cells is being currently designed. In this second generation CAR T cells, PD-1 is disrupted via CRISPR/Cas9 to enhance the anti-tumor activity of GPC3-CAR T cells against HCC [147]^.^ GPC3-CAR co-expressing co-stimulatory molecule ICOSL-41BB promotes CAR T cell proliferation and tumor rejection [148]. NKG2D-based CAR T cells efficiently lysed NKG2DL^+^ HCC cells invitro and potently eradicated NKG2DL^+^HCC xenografts [149]. The IL-12- inducible GPC3-CAR-T cells could broaden the application of CAR-T-based immunotherapy to patients intolerant of cytoreductive chemotherapy and might provide an alternative therapeutic strategy for patients with GPC3^+^cancers [150]. Intratumoral injection of AFP-CAR T cells significantly regressed both HepG2 and AFP_158_-expressing tumors in mouse xenograft models [151]. MUC-1, EpCAM, AFP, and CEA are the targets of CAR T cells with potential utility for treatment of HCC patients [152]. Recently, CAR-T cells targeting PD-L1 were shown to be effective in suppressing tumor growth of human xenograft models of HCC, nonetheless CAR PD-L1 T cells might recognize and kill normal cells expressing PD-L1 and therefore its safety needs to evaluated before clinical application [153].

## 8. Perspectives

Although very few clinical trials are currently exploring NK cells as a therapeutic option for HCC patients, they represent one of the promising candidates in the development of immunotherapies against advanced HCC. The potential advantages that NK cell therapies have over T cell therapies include more manageable safety profiles and no requirement for autologous cells for adoptive transfer. Furthermore, combinatorial strategies are underway to determine whether the addition of tyrosine kinase inhibitors, thermal ablation, anti-VEGF antibody bevacizumab, or additional immune CPIs could augment the proportion of HCC patients responding. The flexible combination of immunotherapy and other complementary therapies might offer the required breakthrough in clinical efficacy of HCC treatment. The application of recently developed high-throughput singe-cell multi-omics techniques to understand the phenotypic and functional composition of the NK cell populations will contribute to unraveling hepatic NK cell diversity and defining effective markers. Innovations in ICIs and NK cell therapeutic can potentially synergize to re-ignite anti-tumor immunity in HCC patients, improve the magnitude and durability of anti-tumor responses and may be a paradigm shift in the treatment for this disease.

## Figures and Tables

**Figure 1 cells-10-01332-f001:**
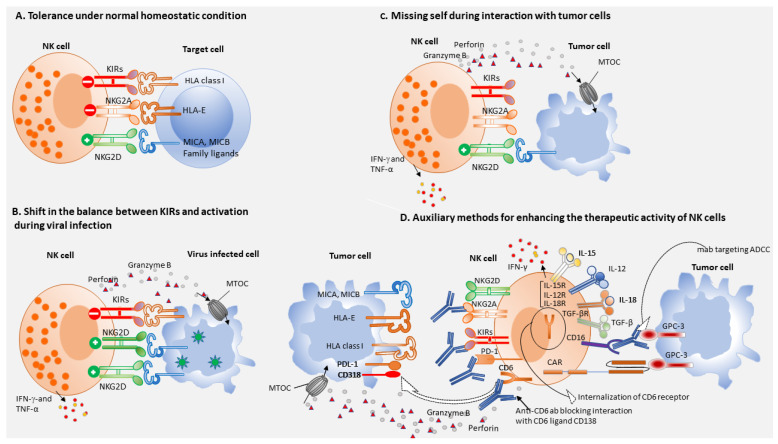
Shift in the balance between inhibitory and activation signaling; auxiliary methods for enhancing the therapeutic efficacy of NK cell-based immunotherapy. NK cells express a broad spectrum of receptors with activating or inhibitory functions and the balance between signaling inputs through these receptors shifts the equilibrium towards tolerance or activation of cytotoxicity against the target cells. (**A**). Under normal homeostatic condition, when the overall inhibitory signaling strength outweighs activating receptor signaling, NK cell activation is aborted resulting in tolerance. (**B**). During viral infection, cells upregulate stimulatory ligands for NK cell activating receptor NKG2D and as a result activation signal surpasses signaling through the inhibitory receptors KIRS and NKG2A, resulting in cytokine release and cytolysis of target cells. (**C**). Class I MHC ligands of NK cell inhibitory receptors are downregulated in tumor cells during malignant transformation and the loss of inhibitory signals causes positive signaling which leads to NK cell activation and tumor cell killing, referred to the “missing-self” phenomena. (**D**). Immune-checkpoint inhibitors could potentially relieve suppression of NK cell-mediated cytotoxicity by preventing inhibitory signaling through PD-1, CD6, NKG2A, and killer immunoglobulin-like receptors (KIRs). In addition, activation of members of the natural cytotoxicity receptor family, such as NKG2D, results in the release of preformed cytolytic granules containing granzyme B and perforin. Pro-inflammatory cytokines such as IL-12 and IL-18 enhance NK cell effector function and cytokine secretion, whereas anti-inflammatory cytokine TGFβ inhibits NK cell function. IL-15 is a crucial homeostatic cytokine for NK cells and exogenous IL-15 can cause NK cell activation. NK cells expressing Fc receptors for antibodies, can recognize and kill sensitized tumor cells through ADCC. Genetically modified CAR NK cells accelerate the antitumor activity of adoptive NK cell therapies.

## Data Availability

Not applicable.

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
