# Peer review of "Natural Killer Cells and T Cells in Hepatocellular Carcinoma and Viral Hepatitis: Current Status and Perspectives for Future Immunotherapeutic Approaches"

_cells, 2021, doi:10.3390/cells10061332_

Round 1

Reviewer 1 Report

This manuscript by Kalathil and Thanavala reviews the current literature on the role of NK and T cells in HCC and viral hepatitis with a particular focus on immunotherapeutic approaches. The manuscript was comprehensive and well written.

This review is therefore interesting and addresses an important and timely topic that has therapeutic implications. Thereby, the main flaw relies on the absence of a specific section on metabolism of T and NK cells which is fundamental for new treatment options for liver tumor and hepatitis. Indeed, the author mentioned “the IL-12-inducible GPC3-CAR-T cells” without any previous description on the role of IL-12 for instance in HBV.

Therefore, I will suggest in order to complete the overview on the future immunotherapeutic approaches, to add some sentences and references related to IL-12 (e.g. Schurich A et al 2013, 2016) and autophagy in HBV (e.g. Swadling L et al Cell Rep 2020 and Acerbi G et al J Hep 2021), to mitochiondrial-dependent energy derangements in HBV and HCV (e.g. Fisicaro P et al Nat Med 2017; Barili et al Nat Comm 2020; Aregay A et al J Hepatol. 2019 Nov;71(5):889-899) with several tested compounds for immune restoration (see also Amin OE et al Hepatology 2020; Bénéchet AP Nature. 2019 Oct;574(7777):200-205).

This topic is mandatory also for NK cells in HCC progression (see for instance Zheng X et al Nat Immunol. 2019 Dec;20(12):1656-1667; C Harmon et al Cancer Immunol Res 2019, pp. 335-346; A Zecca et al Cancer Immunol Immunother 2020).

Very minor points:

- the authors should mention the figure in the correct order (from A to D) in the main text.

-the authors should correct typing mistakes, for instance checkpoint/check-point, IL2 instead of IL-2, CDD49a+ and many others.

Author Response

Response to reviewers’ comments

Reviewer #1. Comments and Suggestions for Authors

This manuscript by Kalathil and Thanavala reviews the current literature on the role of NK and T cells in HCC and viral hepatitis with a particular focus on immunotherapeutic approaches. The manuscript was comprehensive and well written.

Comment-1. This review is therefore interesting and addresses an important and timely topic that has therapeutic implications. Thereby, the main flaw relies on the absence of a specific section on metabolism of T and NK cells which is fundamental for new treatment options for liver tumor and hepatitis. Indeed, the author mentioned “the IL-12-inducible GPC3-CAR-T cells” without any previous description on the role of IL-12 for instance in HBV.

Therefore, I will suggest in order to complete the overview on the future immunotherapeutic approaches, to add some sentences and references related to IL-12 (e.g. Schurich A et al 2013, 2016) and autophagy in HBV (e.g. Swadling L et al Cell Rep 2020 and Acerbi G et al J Hep 2021), to mitochiondrial-dependent energy derangements in HBV and HCV (e.g. Fisicaro P et al Nat Med 2017; Barili et al Nat Comm 2020; Aregay A et al J Hepatol. 2019 Nov;71(5):889-899) with several tested compounds for immune restoration (see also Amin OE et al Hepatology 2020; Bénéchet AP Nature. 2019 Oct;574(7777):200-205).

This topic is mandatory also for NK cells in HCC progression (see for instance Zheng X et al Nat Immunol. 2019 Dec;20(12):1656-1667; C Harmon et al Cancer Immunol Res 2019, pp. 335-346; A Zecca et al Cancer Immunol Immunother 2020).

Response-1. We are most grateful to the reviewer for the recognition that our manuscript was well written and addresses an important and timely topic that has therapeutic implications. Below we have addressed individual comments point by point. We believe the revisions we have made have helped to further enhance our manuscript and we appreciate the issues raised by the reviewer. As advised, all references related to IL-12 and autophagy in HBV and mitochondria-dependent energy derangement in HBV and HCV have been incorporated in an additional paragraph added to this section as follows (Page-4, line 143 to 181).

The metabolic fitness of virus specific CD8+ T cells plays a central role in mounting an efficacious anti-viral immune response during chronic HBV or HCV infection. One important mechanism responsible for the failure of intrahepatic HCV‐specific CD8+ T cells to clear HCV in persistent infection is due to the loss of mitochondrial fitness. In HCV infection, virus-specific CD8+ T cells display a marked upregulation of transcription of genes associated with impaired glycolytic and mitochondrial functions, leading to exhaustion of HCV-specific T cells (Barili et al 2020). HBV-specific CD8+ T cells are markedly impaired at multiple levels and show substantial downregulation of various cellular processes centered on extensive mitochondrial alterations (Fiscaro, P et al 2017). Multiple strategies to increase the mitochondrial potential of exhausted virus specific CD8+ T cells thereby rescuing them from functional exhaustion has been reported recently. In chronic HBV infection, IL-12 has been shown to revitalize exhausted HBV-specific T cells by reducing their dependence on glycolysis and thereby increasing their mitochondrial potential, which is critical for viral control. Furthermore, IL-12 administration has been reported to significantly decrease the levels of pro-apoptotic protein Bim, which potentially causes attenuation of HBV specific CD 8+ T cells (Schurich et al 2013, 2016). IL-15 induced upregulation of autophagy in human liver-resident CD8memory T cells has been described as a protective mechanism against mitochondrial depolarization and to optimize effector T cell function (Swadling L et al 2020). In chronic HBV patients, IL-2 has been described as a potent immunotherapeutic that can rescue CD8+ T cells rendered dysfunctional by intrahepatic priming; this highlights the importance of IL-2-based strategies that should be considered for the treatment of chronic HBV infection (Benechet A, et al 2019). Significant improvement of mitochondrial function with concomitant enhancement in anti-viral CD8+ T cell functions was elicited by mitochondrion-targeted antioxidants, which suggests a central role for reactive oxygen species (ROS) in T cell exhaustion (Fiscaro, P et al 2017). Thus, mitochondria represent promising therapeutic targets to treat chronic HBV infection. Additionally, combinatorial treatment of polyphenolic compounds (resveratrol, oleuropein) and antioxidants improved anti-HBV T cell responses in chronic HBV patients. Therefore, combination of antioxidants and natural polyphenols represents a promising strategy for chronic HBV therapy (Acerbi, G et al 2021). GS-9688 (selgantolimod),  a toll-like receptor 8 (TLR8) agonist induces cytokines in human PBMC that are able to activate antiviral effector function by multiple immune cells, including HBV-specific CD8+ T-cells, T follicular helper (TFh) , NK cells and MAITs (Amin OE et al 2020). In chronic HCV patients treated with direct-acting antiviral (DAA) therapy, exhausted HCV-specific CD8+ T cells remain functionally and metabolically impaired at multiple levels and altered mitochondrial dysfunction exhibited by exhausted HCV-specific CD8+ T cells could not be normalized even after HCV clearance in most patients. This finding implies that HCV cure does not lead to protective immunity and that re-infections have frequently been observed (Aregay A et al 2019). Nevertheless, functional exhaustion of HCV specific CD8+ T cells associated with impairment in glycolytic and mitochondrial function can be alleviated by histone methyltransferase inhibitors (Barili, et al 2020).

Comment-2. This topic is mandatory also for NK cells in HCC progression (see for instance Zheng X et al Nat Immunol. 2019 Dec;20(12):1656-1667; C Harmon et al Cancer Immunol Res 2019, pp. 335-346; A Zecca et al Cancer Immunol Immunother 2020).

Response-2. As suggested, all the references relevant to NK cell mitochondrial fitness in HCC have been cited in an additional paragraph added in this section as follows (Page-7 & 8, line 344 to 362).

Metabolic fitness of NK cells has a crucial role in tumor surveillance. Fragmentation of mitochondria in liver NK cells of HCC patients was correlated with reduced cytotoxicity and NK cell loss, resulting in tumor evasion from NK cell surveillance, with predicted poor survival in HCC patients. Inhibition of mitochondrial fragmentation improved mitochondrial metabolism, survival, and the antitumor capacity of NK cells (Zheng X et al 2019). In colorectal liver metastasis (CRLM), liver-resident NK cells infiltrating into the tumors were unable to regulate intracellular pH resulting in mitochondrial stress and apoptosis. Hence, targeting CRLM metabolism represents a novel therapeutic strategy to restoring local NK-cell activity and preventing tumor growth (Harmon, C et al 2019). Functional rescue of NK-cells in solid tumors represents a central aim for new immunotherapeutic strategies. Metabolic defects and functional impairment of circulating NK-cells in patients with HCC could be attributed to TGF-β. Incubation of NK-cells from healthy donors, with TGF-β-rich plasma from HCC patients can recapitulate the metabolic and functional impairment predicted by phenotypic and gene expression profiles of HCC patients. TGF-β antagonism could partially restore functional defects of NK-cells, suggesting that TGF-β, among other molecules, may represent a suitable target for immunotherapeutic intervention aimed at functional restoration of NK-cell function (Zecca, A et al 2020). These studies highlight the potential mechanism of immune escape through mitochondrial dysfunction that might be targetable and could reinvigorate NK cell-based cancer treatments.

Very minor points:

Comment-1.  The authors should mention the figure in the correct order (from A to D) in the main text.

Response-1. Figure has been mentioned in the main text in the correct order of A to D.

Comment-2.  the authors should correct typing mistakes, for instance checkpoint/check-point, IL2 instead of IL-2, CDD49a+ and many others.

Response-2. Typing mistakes have been fixed

Reviewer 2 Report

In this review by Suresh Gopi Kalathil and Yasmin Thanavala, authors discuss the possible immunotherapeutic strategies by NK- and T-cells in HCC. Literature on these topics is quite wide and it is understandable that not all aspects can be addressed in particular dealing with both NK- and T-cells. A review focused on either one would have allowed more space to go in greater detail, discussing preclinical and clinal studies.

Main issues

- Abstract Line 17: the recent phase 3 trial by Finn RS et al (N Engl J Med. 2020 May 14;382:1894-1905) is now first line treatment for HCC and it is based on anti-PDL-1 in association with anti-VEGF therefore the statement in the abstract that it has been with limited success should be changed

- CAR-NK cells paragraph should explain what is a CAR-NK and FCR. Line 324-328 should be better clarified going more in detail.

- T cells-based immunotherapy of HCC. Discussing genetically modified T-cells for HCC, the work on HBV-specific TCR-redirected T cells (Qasim W, et al. J Hepatol. 2015; 62:486-91) should be quoted because first in human proof of principle of CAR-T for HCC.

- line 353-354 the reader after reading that frequency of TAA-specific T-cell is low expect some additional conclusion.

- line 359-361 t is not clear, why MAIT cells are a therapeutic target and why this is relevant for immunotherapy, go more in detail.

-following paragraph: discussing the principle of synergistic effect that can be achieved by combination of local ablation and immunotherapy; it would be important to quote the study of Duffy AG et al J Hepatol. 2017;66:545-551 that is one of first trial demonstrating proof of principle

- line 377-378 “Effector CD8+ T cells are in an exhausted state …” it is not true or at least this is for sure not one of the most relevant mechanisms among the many mechanisms exploited by the tumor to escape immune surveillance. With some exception TAAs are not determining T-cell exhaustion.

- line 391-394 beside the phase I/II trial there is the phase III trial of Finn RS et al, NEJM 2020.

Minor issues

-Figure 1B is the first described, the figure should follow the order in the manuscript A to D

Typing mistakes

Line 238, 256, 288

Author Response

Reviewer # 2. Comments and Suggestions for Authors

In this review by Suresh Gopi Kalathil and Yasmin Thanavala, authors discuss the possible immunotherapeutic strategies by NK- and T-cells in HCC. Literature on these topics is quite wide and it is understandable that not all aspects can be addressed in particular dealing with both NK- and T-cells. A review focused on either one would have allowed more space to go in greater detail, discussing preclinical and clinal studies.

Main issues

Comment-1. Abstract Line 17: the recent phase 3 trial by Finn RS et al (N Engl J Med. 2020 May 14;382:1894-1905) is now first line treatment for HCC and it is based on anti-PDL-1 in association with anti-VEGF therefore the statement in the abstract that it has been with limited success should be changed.

Response-1. The statement in abstract Line -17 18 has been changed to ‘first line’ treatment and ‘limited success’ has been removed.

Comment-2- CAR-NK cells paragraph should explain what a CAR-NK is and FCR. Line 324-328 should be better clarified going more in detail.

Response-2.  As suggested, CAR-NK cells and FcR have been described in detail as follows. (Page-8, line 372 to 376; line 390 to 397).

CAR-NK cells are primary NK cells genetically engineered to express chimeric antigen receptor (CAR) such as AFP or GPC-3 which facilitate binding of relevant antigen in a specific manner. Additionally, NK cells express receptors for Fc portion of immunoglobulin (FcR) which can bind sensitized target cells and mediate cytolysis of NK cell susceptible target cells (ADCC). Granule polarization is a critical step leading to NK cell mediated cytolysis of resistant NK cell target, MDA-MB-453 breast cancer cells.  Unmodified NK cells create conjugates with these resistant cancer cells and respond by granule clustering but fail to induce granule polarization and consequent release of lytic enzymes. Nevertheless, retargeting by ErbB2-targeted CAR-expressing NK cell or high affinity FcR transgenic NK cells plus monoclonal antibody Herceptin (ADCC) provides the signals necessary to achieve granule polarization and tumor cell lysis, thereby circumventing tumor cell resistance. 

Comment-3. - T cells-based immunotherapy of HCC. Discussing genetically modified T-cells for HCC, the work on HBV-specific TCR-redirected T cells (Qasim W, et al. J Hepatol. 2015; 62:486-91) should be quoted because first in human proof of principle of CAR-T for HCC.

Response-3.  As advised by the reviewer, while discussing ‘genetically modified T cells for HCC’ the work on HBV-specific TCR-redirected T cells by Quasim W, et al 2015 has been cited and the following statement has been incorporated “Genetically modified autologous T cells that express an HBsAg specific T cell receptor, mediated a reduction in HBsAg levels in HBV related HCC without exacerbation of liver inflammation or ‘off-target’ toxicities. This is the first human proof of principle study for TCR redirected therapy against HBV associated hepatic malignancies (Page- 10, line 494 to 497).

Comment-4. line 353-354 the reader after reading that frequency of TAA-specific T-cell is low expect some additional conclusion.

Response-4.  As suggested, an additional sentence on the low frequency of TAA-specific T-cell has been provided as follows: “RFA enhances TAA‐specific T cell responses and the number of T cells induced is associated with recurrence‐free survival of HCC patients. To maintain the TAA‐specific T cell responses induced by RFA and to improve the immunological outcome for HCC, additional treatment by vaccine or immunomodulatory drugs might be useful (Page-9, line 427 to 432).

Comment-5- line 359-361 it is not clear, why MAIT cells are a therapeutic target and why this is relevant for immunotherapy, go more in detail.

Response-5. The relevance of MAIT cells as a therapeutic target for immunotherapy has been discussed in detail as follows “Even though normal mucosal associated invariant T (MAIT) cells could induce apoptosis of HCC cells, they are exhausted and functionally impaired in the tumor environment  In the context of liver TME, tumor-derived MAIT cells secrete less IFN-γ, IL-17 and granzyme B, perforin, but they secrete more IL-8, which promotes tumor angiogenesis and progression (Zhang et al 2020).  Therefore, a high density of tumor infiltrating MAIT cells have been reported to indicate disease progression and poorer outcome in HCC and these cells could potentially serve as a novel therapeutic target in T cell-based immunotherapy of liver cancer (Page- 9, line 436 to 445).

Comment-6. -following paragraph: discussing the principle of synergistic effect that can be achieved by combination of local ablation and immunotherapy; it would be important to quote the study of Duffy AG et al J Hepatol. 2017; 66:545-551 that is one of first trial demonstrating proof of principle

Response-6. We thank the reviewer for pointing out an important reference we missed, while discussing the principles of synergistic effects that can be achieved by the combination of local ablation and immunotherapy in HCC. As advised, the study of Duffy AG et al 2017 was incorporated in the discussion. “Ablative therapies induce a peripheral immune response which may enhance the effect of anti-CTLA-4 treatment in patients with advanced hepatocellular carcinoma. Tremelimumab in combination with tumor ablation represents a potentially novel treatment modality for patients with advanced HCC, that leads to the accumulation of intratumoral CD8+ T cells in conjunction with reduction in HCV viral load” (Duffy, AG et al 2017) (Page- 9, line 454 to 458).

Comment-7. line 377-378 “Effector CD8+ T cells are in an exhausted state …” it is not true or at least this is for sure not one of the most relevant mechanisms among the many mechanisms exploited by the tumor to escape immune surveillance. With some exception TAAs are not determining T-cell exhaustion.

Response-7.  As suggested, the sentence has been modified as “Effector CD8+ T cells are in an exhausted state in advanced HCC patients, caused by persistent antigen stimulation and immunosuppressive factors.  In this context, we would like to point out that PD-1 is an inhibitory checkpoint receptor that is expressed on T cells after chronic antigenic stimulation. Moreover, the combination of atezolizumab (anti-PD-1) and bevacizumab(anti-VEGF) in patients with unresectable HCC (including but not stratifying viral and non-viral etiologies) showed a progression-free survival rate of over 15 months, that is superior compared to that achieved by treatment with the multi-kinase inhibitor sorafenib ( Finn RS et al, NEJM 2020) ( Page-10, line 467, 480 to 483).

Comment-8. - line 391-394 beside the phase I/II trial there is the phase III trial of Finn RS et al, NEJM 2020.

Response-8. Besides the phase 1/II, the reference for phase III trial (Finn RS et al, NEJM 2020, Ref # 141) has been incorporated (Page 10, 480 to 483)

Minor issues

Comment-1. Figure 1B is the first described, the figure should follow the order in the manuscript A to D

Response-1.  Description of figures in the text has been changed in the order of A to D.

Comment- 2. Typing mistakes Line 238, 256, 288

Response-2.   Typing mistakes in Line 238, 256, 288 have been fixed.

Round 2

Reviewer 2 Report

The manuscript has been improved in the revised version. 

Just a comment regarding CD8-cell exhastion. Expression of PD-1 is not necessarily the result of persistent antigen stimulation it can depend on different mechanisms i.e. local release of IFM-gamma etc.